# Ti-40Al-10Nb-10Cr Porous Microfiltration Membrane with Hierarchical Pore Structure for Particulate Matter Capturing from High-Temperature Flue Gas

**DOI:** 10.3390/membranes12020104

**Published:** 2022-01-18

**Authors:** Wanyuan Gui, Zhenjing Shi, Yin Zhang, Yongfeng Liang, Jingyan Qin, Yanli Wang, Junpin Lin, Benli Luan

**Affiliations:** 1National Center for Materials Service Safety, University of Science and Technology Beijing, Beijing 100083, China; guiwy@ustb.edu.cn; 2State Key Laboratory for Advanced Metals and Materials, University of Science and Technology Beijing, Beijing 100083, China; s20201214@xs.ustb.edu.cn (Z.S.); liangyf@skl.ustb.edu.cn (Y.L.); wan-gyl@ustb.edu.cn (Y.W.); 3Xi’an Thermal Power Research Institute Co., Ltd., Xi’an 100083, China; yinzhang330@163.com; 4School of Mechanical Engineering, University of Science and Technology Beijing, Beijing 100083, China; qin-jingyanking@foxmail.com; 5Department of Chemistry, Western University, London, ON N6A 5B7, Canada

**Keywords:** intermetallic, TiAl porous alloys, microfiltration membrane, separation and purification, high-temperature application

## Abstract

TiAl-based porous microfiltration membranes are expected to be the next-generation filtration materials for potential applications in high-temperature flue gas separation in corrosive environments. Unfortunately, the insufficient high-temperature oxidation resistance severely limits their industrial applications. To tackle this issue, a Ti-40Al-10Nb-10Cr porous alloy was fabricated for highly effective high-temperature flue gas purification. Benefited from microstructural changes and the formation of two new phases, the Ti-40Al-10Nb-10Cr porous alloy demonstrated favorable high-temperature anti-oxidation performance with the incorporation of Nb and Cr high-temperature alloying elements. By the separation of a simulated high-temperature flue gas, we achieved an ultra-high PM-removal efficiency (62.242% for PM_<2.5_ and 98.563% for PM_>2.5_). These features, combined with our experimental design strategy, provide a new insight into designing high-temperature TiAl-based porous materials with enhanced performance and durability.

## 1. Introduction

The immense potential in energy conversion and storage, adsorption and separation applications has generated significant interest in the design and synthesis of hierarchically porous materials [1,2,3,4,5]. Hierarchically porous materials have many unique features, such as tunable porous structures, controllable macroscopic morphologies, a large surface area and an easily functionalizable surface, making them some of the most promising engineering structural materials [6,7,8,9,10,11,12,13]. High-temperature flue gases discharged from electric power, petroleum, chemical and metallurgical operations have the characteristics of high temperature (above 800 °C), high oxygen content, high sulfur content, high nitrogen content and large amounts of dust content [7,8,9]. Dust removal from these high-temperature flue gases remains a major challenge due to the formation of blockage caused by particle-containing high-temperature flue gases and the corrosion of dust removal equipment. Porous metals [14,15] and porous ceramics [16,17,18,19] are widely utilized where high-temperature flue gas is initially released to take full advantage of the filtration efficiency. Unfortunately, the poor oxidation resistivity, poor corrosion resistance and intolerance at elevated temperatures of porous metals and the severe brittleness, poor thermal vibration resistance and unworkability of porous ceramics have severally restricted their potential applications in high-temperature flue gas purification. As such, it is of considerable significance to develop functional porous materials for high-temperature flue gas purification with a simple, highly efficient and scalable approach.

TiAl-based porous materials have been very promising candidates as high-temperature structural materials for high-temperature flue gas purification, because they contain a mixture of metallic and covalent bonds that provide sound mechanical properties with outstanding corrosion resistance and excellent oxidation resistance above 600 °C [20,21,22,23]. However, TiAl-based porous materials still need to be improved, owing to the insufficient oxidation resistance in the envisioned application temperature range of 800 °C–1000 °C. The main reason for the inadequate performance is related to the formation of both TiO_2_ and Al_2_O_3_ rather than continuous Al_2_O_3_ during long-term high-temperature oxidation [19,24,25,26]. For example, the oxidation products of a binary γ-TiAl isothermally oxidized at 1000 °C for 48 h include not only Al_2_O_3_ (α-alumina), but also TiO_2_ (rutile titanium dioxide), TiN, Ti_2_AlN and α_2_-Ti_3_Al [27]. More specifically, the oxide scale of TiAl alloys generally consists of three layers, an outer layer of TiO_2_, an intermediate layer of Al_2_O_3_ and a porous inner layer consisting of TiO_2_ and Al_2_O_3_ grains. A great deal of research has already been conducted to enhance the anti-oxidation resistance of TiAl-based alloys above 800 °C through the addition of ternary and quaternary alloy elements into TiAl-based alloys, such as Nb [28], Ta [29], Ni [30], Y [31], B [16], Si [32], W [33] and Mo [34], either to form a protective scale or to slow down the oxygen diffusion rate. However, these studies are still at the experimental stages based on bulk TiAl-based alloys. It is still unclear whether or not the benefits would apply to TiAl-based porous alloys, particularly in controlling the surface morphology and pore parameters. The alloy design should be highly effective in improving high-temperature corrosion resistance without compromising the pore parameters. As such, the development of a simple and effective method to substantially improve the high-temperature oxidation resistance of TiAl-based materials while maintaining the desired porous structures is much needed.

The main objective of this study is to fabricate a novel TiAl-based porous material with the addition of Nb and Cr elements for high-temperature applications. The formation mechanism of the new porous material was investigated and the effects of Nb and Cr doping into the TiAl porous alloy were also demonstrated. Furthermore, the enhancement of anti-oxidation resistance was studied through the characterization analyses before and after high-temperature oxidation at 900 °C for 100 h. This new research project resulted in a new TiAl-based material with good high-temperature oxidation resistance and excellently structural stability for high-temperature PM capturing.

## 2. Experimental Section

### 2.1. Materials

The chemical composition of all samples are as follows: commercial Ti, Al, Nb and Cr powders with a purity of 99.9% and an average particle size of less than 50 μm. All these powders were supplied by DK nano technology Co. Ltd., Beijing, China.

### 2.2. Instruments

The morphological features of the Ti-48Al, Ti-48Al-6Nb, Ti-48Al-2Nb-2Cr and Ti-40Al-10Nb-10Cr porous materials were observed with a field emission scanning electron microscopy (FESEM, ZEISS SUPRA 55, Carl Zeiss, Germany), while the compositional analyses of the samples were performed using energy dispersion spectrometry (EDS). X-ray diffraction (XRD; Multipurpose X-ray Diffractometer TTR III, Rigaku Co., Tokyo, Japan) was used for phase analysis. The pore structure was examined by FESEM and the pore parameters were measured by mercury intrusion porosimetry (MIP; Quantachrome AUTOSCAN-33, Boynton Beach, Florida, USA).

### 2.3. Preparation Process of TiAl-Based Porous Materials

In the preparation process of TiAl-based porous materials, as shown in Figure 1, commercial Ti, Al, Nb and Cr powders with the molar ratios of 52:48, 46:48:6, 48:48:2:2 and 40:40:10:10 were mixed, followed by ball milling at 120 rpm in a ball crusher for 24 h (ball-to-powder weight ratio of 4:1) and the mixtures were subsequently pressed into green pellets with a diameter of 30 mm under the pressure of 230 MPa. A four-step heat-treatment process in a vacuum was then conducted to fabricate TiAl-based porous materials. More specifically, the pellets were heated at 120 °C/1 h for vapor evaporation, at 600 °C/3 h and 900 °C/3 h for the Al and Ti reaction, Al and Nb reaction and phase transformation and, finally, at 1350 °C/3 h to form the TiAl-based porous materials. 

### 2.4. High-Temperature Oxidation of TiAl-Based Porous Materials

To study the thermal cycling oxidation behavior, the Ti-48Al, Ti-48Al-6Nb, Ti-48Al-2Nb-2Cr and Ti-40Al-10Nb-10Cr porous alloys were treated at 900 °C for a total oxidation duration of 100 h. In detail, all the TiAl-based porous materials samples were kept at 900 °C for high-temperature oxidation, removed from the furnace with various oxidation intervals (2 h, 6 h, 10 h, 20 h, 30 h, 40 h, 50 h, 60 h, 70 h, 80 h, 90 h and 100 h), cooled in air (at room temperature) for 1 h for weighing and then placed back in the furnace for continued oxidation for 100 h. Besides weight measurements, FESEM was performed after thermal cycling oxidation treatment.

### 2.5. High-Temperature Filtration Performance of Ti-40Al-10Nb-10Cr Porous Alloy Tests

A home-made high-temperature PM filtration apparatus was applied to evaluate the filtration performance of the Ti-40Al-10Nb-10Cr porous alloys. The PM in the high-temperature PM filtration apparatus was generated by burning incense [35], while the size and concentration of PM before and after filtration were measured by two laser PM sensors (DT9881, CEM). The simulated high-temperature pollutant gas flowing through the Ti-40Al-10Nb-10Cr porous alloys samples (sample specifications: Φ30 × 0.6 mm, with an effective area of about 706.5 mm^2^) was placed inside a quartz tube in a furnace (900 °C, 3000 Pa). Two PM counters were placed at the downstream and upstream of the testing samples, respectively, to measure the PM number before and after filtration. During the experiment, high-temperature PM-containing air flowed at a 2 L/min constant rate through the samples. Three Ti-40Al-10Nb-10Cr porous alloys samples were tested to ensure filtration measurement accuracy. Their removal efficiency was calculated by Equation (1) as follows:η = (1 − ξ_1_/ξ_2_) × 100% (1)
where ξ_1_ and ξ_2_ represent the concentrations of incense PM in the downstream and upstream of the filter, respectively.

## 3. Results and Discussion

### 3.1. Phase Composition and Microstructure of TiAl-Based Porous Materials

Figure 1 depicts the typical TiAl binary phase diagram (dashed line) and preliminary phase diagram of TiAl with 10Nb (solid line), respectively. It can be observed that, with a 10 at% Nb addition, the phase diagram of TiAl changed. The melting point of the TiAl alloy increased by about 80~100 °C; the phase transition point (β/β+α) was reduced by about 50~80 °C and the phase region was enlarged and extended to the high Al region; the α(α/α+γ) transition point decreased by about 30 °C; the β+α/α transition temperature decreased by about 50~100 °C; the α single-phase region was compressed and moved to high Al content; the γ phase region extended to low Al content; the maximum solubility of Nb in the α_2_ and γ phases was about 9.5 at% [36]. In addition, Tang and Shemet suggested that whether addition of Cr does good or harm to a TiAl alloy depends on the amounts used; with less-than-4 at% Cr additions, Cr occupied the position of Ti in TiO_2_ in the form of +3 valence, which was harmful to the oxidation resistance. However, with 8 at%~10 at% Cr additions, Cr could promote the formation of an Al_2_O_3_ film and was beneficial to the improvement of antioxidation properties [37]. Therefore, Ti-40Al-10Nb-10Cr was selected as the composition with consideration of the effects of both Nb and Cr on the oxidation resistance for high-temperature applications. 

The XRD patterns obtained from the surfaces of the Ti-48Al, Ti-48Al-6Nb, Ti-48Al-2Nb-2Cr and Ti-40Al-10Nb-10Cr specimens are shown in Figure 2. The Ti-48Al porous alloys were mainly composed of Ti_3_Al and TiAl. Besides Ti_3_Al and TiAl, two new phases, NbAl_3_ and Nb_2_Al, were detected in the Ti-48Al-6Nb and Ti-48Al-2Nb-2Cr samples. In addition, the new phase B2 was also found in Ti-40Al-10Nb-10Cr porous alloys. These findings are similar to the results of our previous study [23,38].

Figure 3a depicts typical FESEM images of Ti-48Al porous materials. The results show that Ti-48Al porous materials mainly consisted of irregular spherical particles, leading to a hierarchically porous skeleton (funnel-shaped with big pore mouth and small pore throat). There existed a lot of pores with different pore diameters among the Ti-48Al skeletons. Compared with Ti-48Al porous materials, a larger number of white irregular particles appeared on the surface of the Ti-48Al-6Nb, Ti-48Al-2Nb-2Cr and Ti-40Al-10Nb-10Cr porous alloys, due to the presence of high Nb phases (NbAl_3_ and Nb_2_Al) and the B2 phase, as shown in Figure 3b–d. In particular, a larger number of irregular particles appeared around the pores of the Ti-40Al-10Nb-10Cr samples, resulting in the formation of an increasing number of small pores on the hierarchically porous skeleton.

### 3.2. Pore Parameters of TiAl-Based Porous Materials

Although the SEM images clearly showed the presence of pore mouths with, predominantly, the size of 20–50 μm, the size of pore throats was less than 10 μm, which made the MIP method still applicable [39] and the pore diameter distribution of the TiAl-based porous alloys were analyzed accordingly, as depicted in Figure 4. As for the Ti-48Al porous alloy, an average pore diameter of 8.319 μm was observed from a Gaussian distribution, with the peak position of the pore diameter occurring around 9.185 μm, 9.376 μm and 10.248 μm for the Ti-48Al-6Nb, Ti-48Al-2Nb-2Cr and Ti-40Al-10Nb-10Cr samples, respectively. Smaller peaks of pore diameters at 2–5 μm for the Ti-48Al, Ti-48Al-6Nb and Ti-48Al-2Nb-2Cr porous alloys and 0–2 μm for the Ti-40Al-10Nb-10Cr porous alloy were also observed, demonstrating a wide range of pore diameter distribution for TiAl-based porous alloys. In addition, detailed pore parameters of the TiAl-based porous alloys are given in Table 1. Compared to the Ti-48Al porous alloy, the pore area and pore volume of the Ti-48Al-6Nb, Ti-48Al-2Nb-2Cr and Ti-40Al-10Nb-10Cr porous alloy all decreased to a different extent, due to their wider pore diameter distribution. After Nb and Cr additions, the porosity increased, except for the Ti-48Al-2Nb-2Cr porous alloy. In addition, the porous skeletons increased due to an increase in the apparent skeletal density. The appearance of the high Nb phase (NbAl_3_ and Nb_2_Al phase) and B2 phase indicated an increase in the Ti-48Al-6Nb, Ti-48Al-2Nb-2Cr and Ti-40Al-10Nb-10Cr porous alloys’ skeletons. 

### 3.3. High-Temperature Oxidation Performance

Figure 5 shows the effect of Nb and Cr elemental additions on the high-temperature oxidation behavior of TiAl-based porous alloys. As was measured, the Ti-48Al porous alloys showed a weight gain of 27.39 g/m^2^ after a thermal cycling treatment at 900 °C for 100 h. Compared to Ti-48Al porous alloys, all the TiAl-based porous alloys after Nb and Cr addition showed a lower oxidation rate, especially the Ti-40Al-10Nb-10Cr sample, with a minimum weight gain of 6.55 g/m^2^. The great improvement of the high-temperature oxidation resistance achieved through the addition of Nb and Cr powders could be explained by the barriers towards the formation of TiO_2_ due to the presence of high Nb phases (NbAl_3_ and NbAl_2_) and B2 phases. More specifically, the presence of the high Nb phases (NbAl_3_ and Nb_2_Al) and B2 phases could act as diffusion barriers, inhibiting the O inward diffusion and the Ti outward diffusion.

Figure 6 shows the micro-pore changes in the TiAl-based porous alloys after 900 °C/100 h isothermal treatment. The Kirkendall voids were quickly removed due to the formation of irregular white TiO_2_ on the surface of the treated Ti-48Al porous alloy, as shown in Figure 6a. Moreover, microcracks could be seen on the surface of the treated Ti-48Al-6Nb porous alloy, attributed possibly to the stress of the high degree of oxidation, as shown in Figure 6b. As for the treated Ti-48Al-2Nb-2Cr and Ti-40Al-10Nb-10Cr samples, little change was observable to its surface after 900 °C/100 h thermal cycling treatment, as shown in Figure 6c,d. The surface compositions of TiAl-based porous alloys after thermal cycling treatment are listed in Table 2. In comparison with the treated Ti-48Al porous alloys, the content of O reduced greatly, while Ti, Al and Nb increased. The results further suggest that, for the Ti-48Al-6Nb, Ti-48Al-2Nb-2Cr and Ti-40Al-10Nb-10Cr porous alloys, the formation of high Nb phases (NbAl_3_ and NbAl_2_) and B2 phases prevented the further formation of TiO_2_ and Al_2_O_3_ during thermal cycling treatment. These results are consistent with their high-temperature oxidation results shown in Figure 5.

The high-temperature PM filtration of Ti-40Al-10Nb-10Cr porous alloys was tested in the device shown in Figure 7a, with high-temperature PM (including PM_<2.5_, PM_>2.5_) filtered through Ti-40Al-10Nb-10Cr porous alloy. The PM_<2.5_ and PM_>2.5_ concentration after filtration using Ti-40Al-10Nb-10Cr porous alloys’ filtration was much lower than the concentration before filtration (Figure 7b,c). The results shown in Figure 7d confirm that both high-temperature PM_<2.5μm_ and PM_>2.5μm_ could be filtered through the Ti-40Al-10Nb-10Cr membrane with a separation efficiency of 62.242% (SD: ±1.099%) and 98.563% (SD: ±0.449%), respectively. Furthermore, a comparison between Ti-40Al-10Nb-10Cr porous alloys and various porous materials in previous studies [2,19,40] shows that the Ti-40Al-10Nb-10Cr sample exhibited a relatively higher PM_>2.5μm_ removal efficiency at a much higher pressure. It is of note that the upper limit of service temperature of as-prepared Ti-40Al-10Nb-10Cr porous alloys could survive a temperature of up to 900 °C. These results indicate that our Ti-40Al-10Nb-10Cr porous alloys could achieve flow-through filtration with high removal efficiency, showing great commercialization prospects for high-temperature PM filtration.

## 4. Conclusions

In conclusion, a simple and effective strategy is proposed for enhancing high-temperature oxidation resistance and corrosion resistance of TiAl-based porous materials. A novel Ti-40Al-10Nb-10Cr porous alloy with controlled lamellar microstructure was fabricated by manipulating a chemical reaction of Ti and Al with Nb and Cr powders. Compared with the Ti-48Al, Ti-48Al-6Nb and Ti-48Al-2Nb-2Cr porous materials, the Ti-40Al-10Nb-10Cr porous alloy exhibited improved high-temperature oxidation resistance (only 6.55 g/m^2^ weight gain after thermal cycling treatment at 900 °C/100 h). This indicates that the Ti-40Al-10Nb-10Cr porous alloy could provide expanded opportunities for higher-temperature PM capturing with ultra-high PM removal efficiencies (98.563% for PM_>2.5_, 62.242% for PM_<2.5_). These findings represent an important step toward fabricating TiAl-based porous alloys using powder metallurgy, achieving an excellent high-temperature oxidation resistance due to the unique structure and demonstrating great potential for applications in environment-related fields where highly effective and robust high-temperature filtration is required.

## Figures and Tables

**Figure 1 membranes-12-00104-f001:**
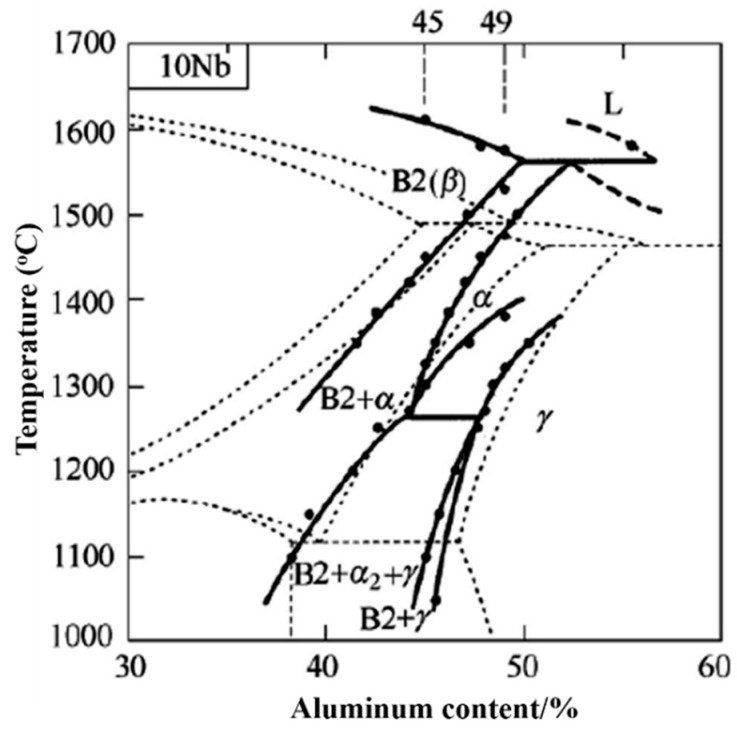
Phase diagram of TiAl binary phase diagram (dashed line) and Ti-40Al alloy with 10Nb (solid line) [36].

**Figure 2 membranes-12-00104-f002:**
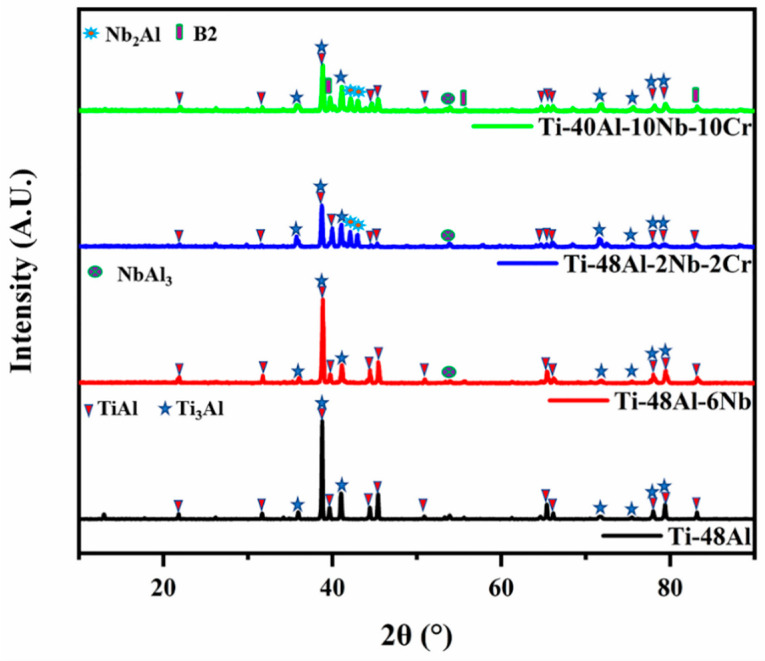
XRD patterns of TiAl-based porous materials: Ti-48Al (black), Ti-48Al-6Nb (red), Ti-48Al-2Nb-2Cr (blue) and Ti-40Al-10Nb-10Cr (green).

**Figure 3 membranes-12-00104-f003:**
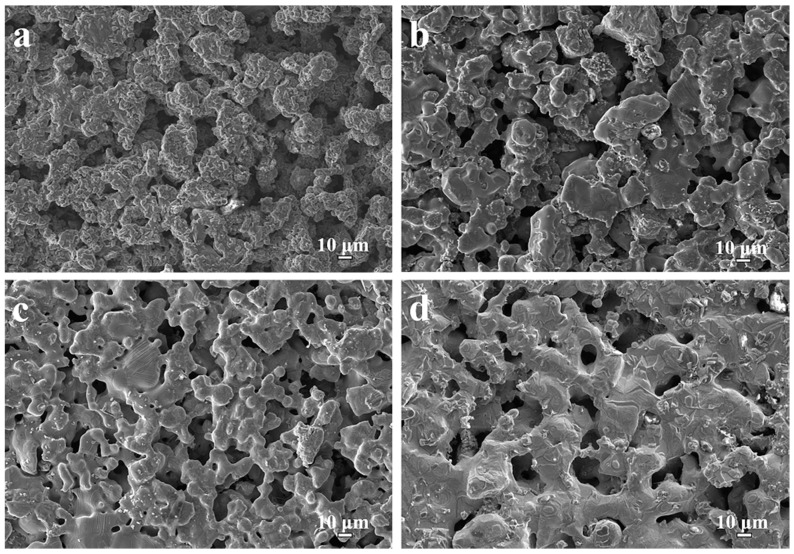
Skeleton surface morphologies of TiAl-based porous alloys: (**a**) Ti-48Al, (**b**) Ti-48Al-6Nb, (**c**) Ti-48Al-2Nb-2Cr and (**d**) Ti-40Al-10Nb-10Cr.

**Figure 4 membranes-12-00104-f004:**
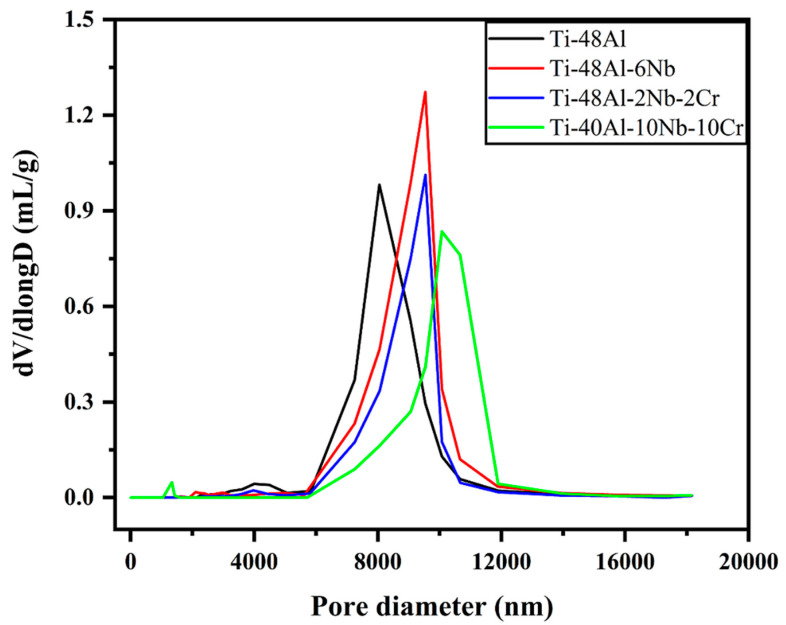
Pore diameter distribution curves of TiAl-based porous materials: Ti-48Al (black), Ti-48Al-6Nb (red), Ti-48Al-2Nb-2Cr (blue) and Ti-40Al-10Nb-10Cr (green).

**Figure 5 membranes-12-00104-f005:**
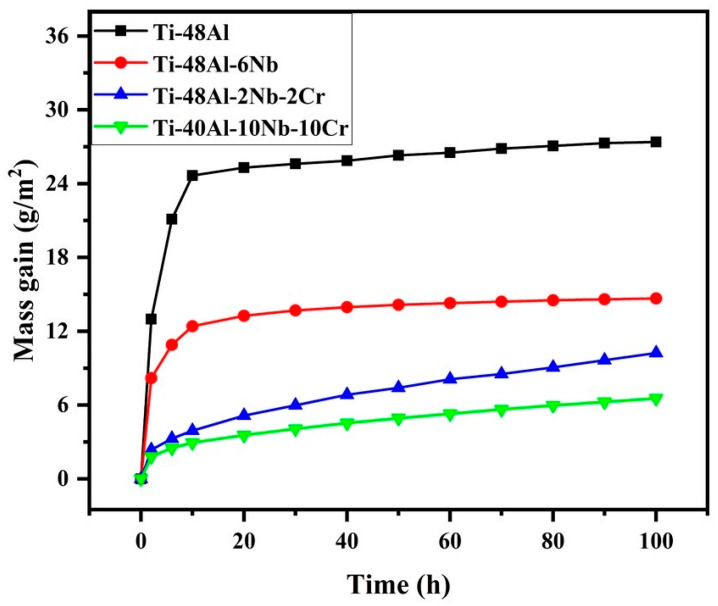
High-temperature oxidation behavior of TiAl-based porous materials: Ti-48Al (black), Ti-48Al-6Nb (red), Ti-48Al-2Nb-2Cr (blue) and Ti-40Al-10Nb-10Cr (green).

**Figure 6 membranes-12-00104-f006:**
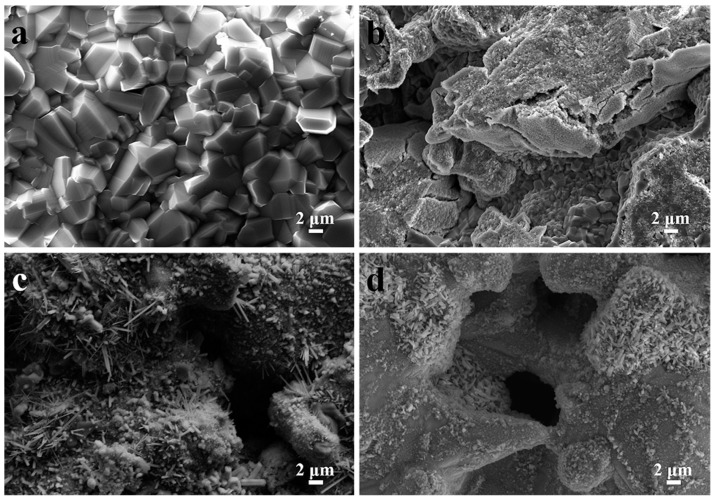
The SEM images of TiAl-based porous alloys after 900 °C/100 h thermal cycling treatment: (**a**) Ti-48Al, (**b**) Ti-48Al-6Nb, (**c**) Ti-48Al-2Nb-2Cr and (**d**) Ti-40Al-10Nb-10Cr.

**Figure 7 membranes-12-00104-f007:**
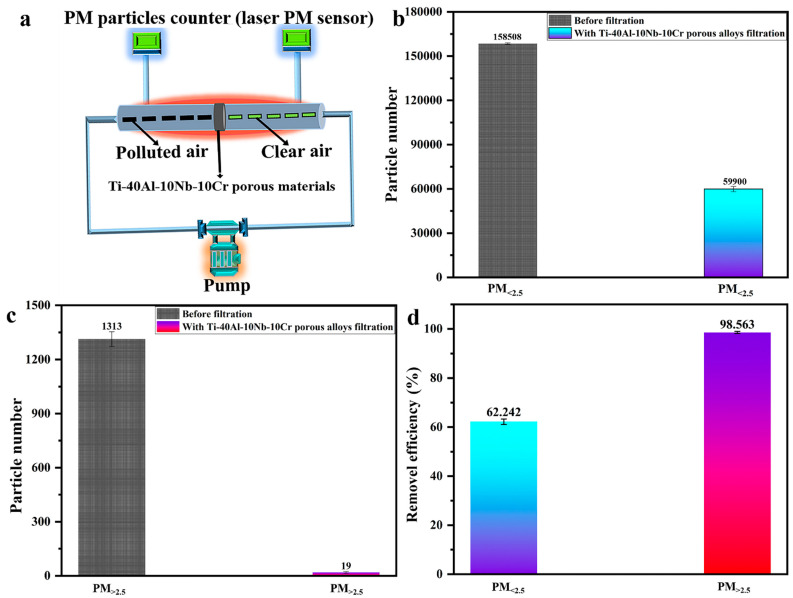
(**a**) Schematic of high-temperature PM filtration setup. (**b**,**c**) PM_<2.5_ and PM_>2.5_ concentration of dust-laden smoke (original) and after filtration under test condition. (**d**) Removal efficiency of Ti-40Al-10Nb-10Cr porous alloys after 60 min filtration test.

**Table 1 membranes-12-00104-t001:** Pore parameters of the TiAl-based porous materials.

Samples	Total Pore Volume (cm^3^/g)	Total Pore Area (m^2^/g)	Bulk Density (g/cm^3^)	Porosity (%)
Ti-48Al	0.113 ± 0.008	0.055 ± 0.007	3.683 ± 0.122	29.445 ± 0.054
Ti-48Al-6Nb	0.118 ± 0.013	0.052 ± 0.011	3.749 ± 0.134	30.577 ± 0.069
Ti-48Al-2Nb-2Cr	0.082 ± 0.005	0.038 ± 0.007	4.644 ± 0.095	27.551 ± 0.038
Ti-40Al-10Nb-10Cr	0.097 ± 0.003	0.040 ± 0.001	4.358 ± 0.037	29.805 ± 0.003

**Table 2 membranes-12-00104-t002:** EDS composition analysis of the surface scans of TiAl-based porous alloys after the thermal cycling treatment shown in Figure 5.

Samples	Ti	Al	Nb	Cr	O
Ti-48Al	32.96	0	0	0	67.04
Ti-48Al-6Nb	34.18	6.51	1.59	0	57.72
Ti-48Al-2Nb-2Cr	39.78	8.45	1.23	0.31	50.23
Ti-40Al-10Nb-10Cr	41.19	11.91	2.05	0.63	44.22

## Data Availability

The data that support the findings of this study are available from the first corresponding author upon reasonable request.

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
