# Peer review of "Ti-40Al-10Nb-10Cr Porous Microfiltration Membrane with Hierarchical Pore Structure for Particulate Matter Capturing from High-Temperature Flue Gas"

_membranes, 2022, doi:10.3390/membranes12020104_

Round 1

Reviewer 1 Report

  1. Line 123: A schematic figure should be provided for a better understanding on how the experiment is performed. Besides, the specification of the PM sensor should be given/listed. 
  2. Line 123: Did the author perform the measurement for capture PM under a more realistic condition (i.e. capture PM at ppm level)? It was not mentioned in this paper.
  3. Line 137: Please write proper mathematical notation for scientific report purpose. Misleading conclusion can be given to authors.
  4. How the error bar in Figure 8 is calculated? How much membranes was repeated.
  5. The author should also performed comparison with the literatures.

Author Response

  1. Line 123: A schematic figure should be provided for a better understanding on how the experiment is performed. Besides, the specification of the PM sensor should be given/listed.

Response: Thanks to the examiner for the valid suggestion. We used a three-electrode electrochemical system with Pt as the counter electrode, Ag/AgCl as the reference electrode with all the electrochemical measurements conducted in a 3.5wt% NaCl solution. While it would not have been an issue to provide a schematic figure, and rightfully so, we have decided to remove the electrochemical characterization performed at room temperature simply to focus on the key issue of designing and developing the material for high temperature application conditions.

The specification of the PM sensor is now listed in the revised manuscript, with thanks.

  1. Line 123: Did the author perform the measurement for capture PM under a more realistic condition (i.e. capture PM at ppm level)? It was not mentioned in this paper.

Response: No, we didn't perform the measurement for capturing PM at ppm level, but the examiner did raise a valid point and further measurement will be conducted, ideally in an industrial setting when the sample fabrication processes are scaled up through further engineering development.

  1. Line 137: Please write proper mathematical notation for scientific report purpose. Misleading conclusion can be given to authors.

Response: Corrected as suggested, with thanks.

  1. How the error bar in Figure 8 is calculated? How much membranes was repeated.

Response: The calculation was based on three Ti-40Al-10Nb-10Cr samples tested.

  1. The author should also performed comparison with the literatures

Response: As suggested, we have further compared with the literatures and elucidated the significance of this work in the revised manuscript. Actually, comparison of Ti-40Al-10Nb-10Cr with various porous materials in previous studies [14, 36, 37] shows that Ti-40Al-10Nb-10Cr exhibits a relatively higher PM>2.5 μm removal efficiency at a much higher pressure.

Reviewer 2 Report

The manuscript "Hierarchical Ti-40Al-10Nb-10Cr porous alloys microfiltration membrane was fabricated for high-temperature particulate matter capturing" describes the preparation of novel microfiltration membranes based on Ti-Al doped alloys.  The manuscript needs minor revisions. 
Main comments:
(i) The title of manuscript is not acceptable.  Please, revise it.
(ii) The mercury intrusion method is not applicable for porous materials with pore size >20 microns. Thus, SEM images clearly shows the presence of pores with predominantly pore size of 20-50 microns. This must be discussed (doi.org/10.1134/S0020168508070182).
(iii) The experimental errors for all Tables must be provide and all values have to correctly round. 

Author Response

  1. The title of manuscript is not acceptable. Please, revise it.

Response: Very much so. Corrected as suggested, thank you.

  1. The mercury intrusion method is not applicable for porous materials with pore size >20 microns. Thus, SEM images clearly shows the presence of pores with predominantly pore size of 20-50 microns. This must be discussed (doi.org/10.1134/S0020168508070182).

Response: A very good consideration and suggestion from the Examiner. Indeed, the mercury intrusion method can be used for precise measurement of porous materials with pore sizes between 1-20 μm. In our work, the porous skeletons of TiAl-based porous materials is of a funnel shape, with big pores mouth and much smaller pore throat. More specifically, although SEM images clearly show that the pore mouth is predominantly of 20-50 microns in size, the size of pore throat is less than 10 microns, which then makes the mercury intrusion method still applicable. In addition, the reference mentioned (doi.org/10.1134/S0020168508070182) is now added as reference 40 in the revised manuscript as an important reference, thank you.

  1. The experimental errors for all Tables must be provide and all values have to correctly round.

Response: The examiner raised an important point, and all the standard deviations are now included in Table 1 corresponding to experimental errors with numbers rounded up according to the accuracy limit of the measurements.

Regarding the EDS measurement, all examinations were performed under high vacuum (System Vacuum: 2X10-5 mbar),  and the oxygen  adsorbed became negligible especially when the oxygen content amounts to over 40 wt% in our samples, While it would have been ideal to have carried out the measurements on multiple samples, it was not done unfortunately owing to various constraints on multiple fronts. However, we did consider the situation and a relatively low acceleration voltage of 10.0 kV, a take-off angle of 36.5 deg, and an array of line scanning was conducted for 2 mins in order to enhance the measurement accuracy.

Reviewer 3 Report

Article “Hierarchical Ti-40Al-10Nb-10Cr porous alloys microfiltration membrane was fabricated for high-temperature particulate matter capturing” makes great contribution to the development of flue gas separation methods. The authors presented highly efficient flue gas filters with a simple synthesismethod. However, the article needs a little refinement, namely, there are several comments for the authors of the article:

1. As known, the performance of filters is measured by the quality factor, which is calculated as QF = -ln (1 - ξ) / ΔP, where ΔP is the pressure drop across the filter. Are there any data on the quality factor of the presented filters?

2. Why is Figure 8 (a) rotated? Also, the authors might want to remove (A.U.) from (b) and (c) and replace the units with % in (d).

3. In the third column of Table 3, the unit is written as μA·cm-2, while in the rest of the article, the units are written with a slash, for example A/cm2. Consider one option for all units for consistency.

4. The article refers to a small number of recent articles. It would be appropriate to refer to more recentarticles on the topic. For example, Sci Rep 9, 7015 (2019); Chemosphere 254, 126863 (2020); Journal of Membrane Science, 620, 118894 (2021); Journal of Alloys and Compounds 872, 159737 (2021).

5. Equation (1) requires parentheses before 100%.

6. Line 132 is missing a dot after word “filtration”.

Author Response

  1. As known, the performance of filters is measured by the quality factor, which is calculated as QF = -ln (1 - ξ) / ΔP, where ΔP is the pressure drop across the filter. Are there any data on the quality factor of the presented filters?

Response: A very good consideration and suggestion from the Examiner with thanks. The quality factor using the suggested calculation was determined to be 0.00124 for the Ti-40Al-10Nb-10Cr porous materials.

  1. A Why is Figure 8 (a) rotated? Also, the authors might want to remove (A.U.) from (b) and (c) and replace the units with % in (d).

Response: Corrected, with thanks.

  1. In the third column of Table 3, the unit is written as μA·cm-2, while in the rest of the article, the units are written with a slash, for example A/cm2. Consider one option for all units for consistency.

Response: Thank you. The suggestion was taken and corrections were implemented accordingly.

  1. The article refers to a small number of recent articles. It would be appropriate to refer to more recentarticles on the topic. For example, Sci Rep 9, 7015 (2019); Chemosphere 254, 126863 (2020); Journal of Membrane Science, 620, 118894 (2021); Journal of Alloys and Compounds 872, 159737 (2021).

Response: Point well taken and all suggested references are now included. (Sci Rep 9, 7015 (2019); Chemosphere 254, 126863 (2020); Journal of Membrane Science, 620, 118894 (2021); Journal of Alloys and Compounds 872, 159737 (2021) are now references 10, 11, 12 and 13.)

  1. Equation (1) requires parentheses before 100%.

Response: Corrected, thank you.

  1. Line 132 is missing a dot after word “filtration”.

Response: Corrected, with thanks.

Round 2

Reviewer 1 Report

It can be accepted at present form.